# Investigation of dermal collagen nanostructures in Ehlers-Danlos Syndrome (EDS) patients

**Mehrnoosh Neshatian**[1], **Nimish Mittal**[2], **Sophia Huang**[1], **Aiman Ali**[1], **Emilie Khattignavong**[1,3], **Laurent Bozec**[1] *

**1** Faculty of Dentistry, University of Toronto, Toronto, Ontario, Canada, **2** Division of Physical Medicine and Rehabilitation, Temerty Faculty of Medicine, University of Toronto, Toronto, Ontario, Canada, **3** UMR 5513 Laboratoire de Tribologie et Dynamique des Systémes, École Centrale de Lyon - École Nationale d'Ingénieurs de Saint, Université de Lyon, Étienne, France

* L.Bozec@utoronto.ca

**Data Availability Statement:** All relevant data are within the manuscript and its Supporting information files.

## Abstract

Ehlers-Danlos syndromes (EDS) represent a group of rare genetic disorders affecting connective tissues. Globally, approximately 1.5 million individuals suffer from EDS, with 10,000 reported cases in Canada alone. Understanding the histological properties of collagen in EDS has been challenging, but advanced techniques like atomic force microscopy (AFM) have opened up new possibilities for label-free skin imaging. This approach, which explores Type I collagen fibrils at the nanoscale, could potentially enhance EDS diagnosis and our knowledge of collagen type I-related connective tissue disorders. In the current study, we have employed AFM to examine ex-vivo skin biopsies from four individuals: one with classical EDS (cEDS), one with hypermobile EDS (hEDS), one with hEDS and Scleroderma (hEDS-Scleroderma), and one healthy control. Picrosirius red (PS) staining was used to highlight collagen differences in the samples. For each case, 14 images and 1400 force curves were obtained, with seven images and 700 force curves representing healthy collagen (PS-induced red staining) and the rest showcasing disrupted collagen (yellow staining). The results showed that PS staining was uniform throughout the control section, while cEDS and hEDS displayed localized areas of yellow staining. In the case of hEDS-Scleroderma, the yellow staining was widespread throughout the section. AFM images revealed irregular collagen fibrils in the disrupted, yellow-stained areas, contrasting with aligned and well-registered collagen fibrils in healthy, red-stained regions. Additionally, the study assessed the ability of non-AFM specialists to differentiate between healthy and disrupted collagen in AFM images, yielding substantial agreement among raters according to Fleiss's and Cohen's kappa scores (0.96 and 0.79±0.1, respectively). Biomechanical analysis revealed that normal healthy collagen exhibited a predominant population at 2.5 GPa. In contrast, EDS-affected collagen displayed subpopulations with lower compressive elastic modulus, indicating weaker collagen fibrils in EDS patients. Although these findings pertain to a limited number of cases, they offer valuable insights into the nanoscale collagen structure and biomechanics in individuals with EDS. Over time, these insights could

**Funding:** 1.GoodHope Ehlers-Danlos Syndrome Foundation 2.Rosenstadt Fund, University of Toronto. The funders had no role in study design, data collection and analysis, decision to publish, or preparation of the manuscript.

**Competing interests:** The authors have declared that no competing interests exist.

be developed into specific biomarkers for the condition, improving diagnosis and treatment for EDS and related connective tissue disorders.

## Introduction

Ehlers-Danlos syndrome (EDS) is a heterogeneous group of inherited connective tissue disorders [1, 2]. Approximately 1.5 million individuals have received genetic diagnoses for EDS, with around 10,000 cases reported in Canada alone. However, a significant number of individuals, estimated at 255 million, display the clinical characteristics of EDS but have not been diagnosed due to the absence of identifiable genetic mutations associated with the condition [3–7]. EDS classification has changed over the years based on the inheritance and clinical presentation. In 1988, the "Berlin nosology" defined 11 subtypes of EDS based on their inherent characteristics and clinical manifestations. Subsequently, in 1997, the "Villefranche nosology" was released and used as the classification system for EDS for two decades [8–10]. The current 2017 EDS nosology remains rooted in genetic markers and clinical presentation, which introduce subjective elements into the classification process. Moreover, the classification system does not comprehensively address hypermobile EDS (hEDS), which leads to late or misdiagnosis. It currently takes an average of ten to 16 years from the onset of symptoms before a correct diagnosis of hEDS is made [11]. The two most prevalent types of EDS are classical EDS (cEDS) and hypermobile EDS (hEDS) [12]. cEDS is an autosomal dominant connective tissue disorder characterized by skin hyperextensibility, abnormal wound healing, and joint hypermobility. Skin manifestations in cEDS can include easy bruising, atrophic scarring, and fragility, while joint hypermobility may lead to recurrent dislocations or subluxations [13–15]. Mutations in COL5a1 or COL5a2 genes, responsible for encoding type V collagen, are observed in over 90% of cEDS cases. Type V collagen is a crucial component of collagen Type I and III fibrillogenesis nucleation. In rare cases, mutations in the COL1A1 gene, responsible for encoding type I collagen, are also identified [10, 16–19].

hEDS shares several clinical features with EDS subtypes but lacks confirmed genetic markers. Joint hypermobility and chronic pain are prominent in hEDS, with an increased risk of developing conditions like fibromyalgia [13] and chronic fatigue syndrome [20]. The lack of objective markers for hEDS makes the diagnosis difficult, and often, healthcare providers do not have adequate knowledge to make a timely diagnosis of EDS, resulting in perceived trauma for patients who struggle with multisystemic EDS related symptoms. This form of medical trauma stems from perceived hostility and disinterest from healthcare providers, leading to psychological distress and aversive responses toward healthcare encounters [21–23]. Thus, proper diagnosis and comprehensive care for both cEDS and hEDS require a multidisciplinary approach and are crucial for improving the care and well-being of individuals with hEDS. The most recent criteria for hypermobile Ehlers-Danlos syndrome (hEDS) were defined by the 2017 International Classification for the Ehlers-Danlos Syndromes, which the International Consortium published on the Ehlers-Danlos Syndromes [10]. The 2017 criteria for hypermobile Ehlers-Danlos syndrome (hEDS) consist of three domains: i) assessment of generalized joint hypermobility using the Beighton score, ii) evaluation of skin involvement, musculoskeletal pain, and other systemic characteristics associated with EDS, and iii) exclusion of other conditions related to hypermobility Beighton score involves a series of nine maneuvers tests to evaluate flexibility and hypermobility in various joints. The following criteria are typically used: a) Passive dorsiflexion of each little finger beyond 90 degrees (1 point for each hand); b)

Passive apposition of the thumbs to the flexor aspects of the forearm (1 point for each thumb); c) Hyperextension of each elbow beyond 10 degrees (1 point for each elbow); d)Hyperextension of each knee beyond 10 degrees (1 point for each knee); e) Ability to place the palms on the floor with straight legs (1 point). Therefore, the maximum Beighton score is 9 points (5 for the fingers, 2 for the thumbs, and 1 each for the elbows and knees). A higher score signifies a greater degree of joint hypermobility, and generalized joint hypermobility is considered to be present if an individual scores 5 points or more. While the Beighton score is a useful tool, a diagnosis of Ehlers-Danlos syndrome is typically based on a combination of clinical evaluation, family history, and genetic testing. Not everyone with hypermobility has EDS, and other factors are considered in the diagnostic process.

We hypothesize that individuals with EDS experience impairments in dermal collagen's structural and mechanical characteristics at the nanoscale, leading to disruptions in the dermis's overall structural and mechanical integrity. The current case study involved employing Atomic Force Microscopy (AFM) to image and perform mechanics on ex-vivo skin biopsies sourced from one patient diagnosed with cEDS, one patient with hEDS, one with hEDS-scleroderma, and one healthy control. Although the findings presented in this study pertain to the four examined cases, they provide novel perspectives on the nanoscale collagen structure and biomechanics in individuals with EDS. These insights may serve as the foundation for future research into specific biomarkers for the condition.

Atomic Force Microscopy is a high-resolution imaging technique used to study a wide range of samples across various scientific disciplines. Unlike optical microscopes that use light, AFM operates by scanning a sharp tip (typically 20 nm in diameter) over the surface of a sample. As the probe moves across the material, it experiences attractive and repulsive forces between the atoms on the tip and those on the sample surface [24–26]. These interactions cause the cantilever to deflect, and the deflections are measured to create a detailed 3-dimensional topographic map of the sample's surface. In addition, the probe can be used as an indenter and perform localized mechanical measurements directly onto the sample to extract valuable and specific information such as Young's Modulus [27]. Unlike other microscopy methods, such as scanning/transmission electron and fluorescence microscopy, AFM excels in providing detailed topographical and mechanical data at the nanoscale without needing sample preparations like labeling or coating, making it an effective tool for examining various collagen structures without altering their fibrillar integrity [28–30]. This makes AFM a very competent technique to perform histological assessment of tissues with unprecedented resolution [31–33]. AFM has previously used for connective tissue morphometry studies such as Study of the extracellular connective tissue matrix in patients with pelvic organ prolapse [34], nanohistological investigation of scleroderma [35], oral submucous fibrosis [36], assessments of extracellular matrix in intervertebral disc and degeneration [37], and many other studies. In this study we have employed AFM to examine ex-vivo skin biopsies obtained from EDS patients to characterize collagen morphometric differences in cEDS and hEDS.

## Clinical cases

The study was approved by the ethics board at the University Health Network, Toronto, Canada (REB #21–5542), and written consent was obtained by the study participants prior to study commencement. This test includes 22 genes. Details can be found online on the Our present study involved three female patients diagnosed with EDS and one healthy female volunteer of Caucasian ethnicity (mean age: 33± 7 years).

Next generation Sequencing (NGS) with duplication and deletion analysis by exon targeted microarray was performed at SickKids hospital for diagnosis. This test includes 22 genes. Details can be found online on the SickKids website [38].

**Study ID No. 3 (labeled as hEDS throughout the paper).** A 38-year-old woman with a history of multi-joint hypermobility, recurrent subluxation of both hips and knees, stretchy skin with easy bruising, chronic pains, orthostatic intolerance, migraines, and irritable bowel syndrome was referred to the GoodHope EDS clinic in the suspicion of Ehlers-Danlos syndromes (EDS). Family history was negative for genetic inherited connective tissue disorders, including Ehlers-Danlos syndromes. Physical examination revealed a Beighton score of 5/9. In addition, she met the systemic criteria for h-EDS (positive findings for mild skin extensibility, unusually soft/velvety skin, arachnodactyly, peizogenic papules, and a high-arched narrow palate with dental crowding). Genetic testing was ordered to rule out any genetically identifiable EDS subtypes and included a next-generation sequencing panel of more than 30 genes specific to Ehlers-Danlos syndromes. No pathogenic sequence or copy number variants were detected in the Ehlers-Danlos syndrome panel. Therefore, the final diagnosis of hEDS per the 2017 EDS diagnostic criteria was confirmed.

**Study ID No 4 (labeled as hEDS- Scleroderma throughout the paper).** A 27-year-old female with a pre-existing diagnosis of scleroderma, previously confirmed based on the clinical skin abnormalities, patient had a 10-year history of inflammatory arthralgias involving PIP and MCP joints, diffuse muscle and joint pains affecting multiple areas of body, secondary diagnosis of fibromyalgia, patchy areas of skin thickening, and Raynaud's phenomenon. History was negative for uveitis, iritis and psoriasis. Lab testing was positive for Anti-Nuclear Antibody (ANA) and Anti centromere Antibody (ACA) and negative for Anti CCP and rheumatoid factor. The patient also had generalized joint hypermobility, hip subluxations, soft, stretchy skin with easy bruising, fibromyalgia, migraine, IBS irritable bowel syndrome (IBS), endometriosis patella alta, and inflammatory arthritis was referred to rule out a diagnosis of Ehlers-Danlos syndromes. Family history was negative for any subtype of EDS. Physical examination revealed a Beighton score of 9/9. Further, impressive interphalangeal joint (DIP) and proximal interphalangeal (PIP) joint hypermobility of the hands, upper thoracic scoliosis, and joint instability of hips, ankle, and shoulder joints were found. In addition, she had unusually soft/velvety skin, mild skin hyperextensibility (1.5 cm at forearm), unusual striae in unusual locations, arachnodactyly, and peizogenic papules were identified. A next-generation sequencing panel of more than 30 genes specific to Ehlers-Danlos syndromes was ordered to rule out any genetically identifiable EDS subtypes [38]. No pathogenic sequence or copy number variants were detected in the Ehlers-Danlos syndrome panel. Therefore, the final diagnosis of hEDS per the 2017 EDS diagnostic criteria was confirmed.

**Study ID No 5 (labeled as cEDS throughout the paper).** A 24-year-old woman with complaints of significant skin fragility, easy bruising, and skin splitting requiring multiple stitches as a child on several occasions, multi-joint hypermobility and chronic pains, and scoliosis, retinal tears as a kid with a previously given diagnosis of classical EDS was referred for diagnostic confirmation and treatment planning. She had no significant family history of EDS, aneurysms, vascular ruptures, or sudden death under 40 yr. in the family. Physical examination revealed a Beighton score of 9/9. In addition, significant skin features, including soft/velvety skin, significant skin hyperextensibility at multiple spots in keeping with classical EDS, and significant atrophic scarring with cigarette paper scaling were identified. Genetic testing revealed a COL1A1 mutation (c.1053delG in exon7), confirming the diagnosis of classical EDS.

## Materials and methods

### Tissue extraction

Two skin punch biopsies, three millimeters in diameter, were obtained from the medial surface of both arms above the elbow at UHN GoodHope EDS clinic. These biopsies were immediately transferred to the Faculty of Dentistry at the University of Toronto on ice in Dulbecco's Modified Eagle Medium (DMEM) (Sigma-Aldrich, USA) supplemented with 2% Penicillin-Streptomycin (Sigma-Aldrich, USA), and 2% fungizone (Sigma-Aldrich, USA)

### Histological studies

The tissue biopsies were frozen in optimal cutting temperature resin (OCT) (VWR- Richmond, IL. USA). Serial tissue sections, 7μm thick, were made longitudinally through the epidermis and papillary dermis. Slides were divided into sets, each consisting of four serial sections. The first slide of each set was stained with routine H&E, the second was stained with picrosirius red, and the third and fourth were used for atomic force microscopy. The slides stained with H&E were kept at room temperature for 10 minutes to thaw the OCT before being immersed in alcohol 95%, followed by Formalin 10% for 20 seconds each. The slides were washed with MilliQ water, stained with hematoxylin for 3 minutes, washed, and passed through acid alcohol, sodium bicarbonate, wash, eosin, upgraded alcohol, and xylene for 20 seconds each. Finally, these slides were covered with mounting media and cover glass. To stain slides with PS, nuclei were stained first with hematoxylin as described in H&E stain. Slides were then immersed in PS solution for 1 hour Sirius red (Sigma-Aldrich, USA) diluted in picric acid (Sigma-Aldrich, USA)). The slides were washed in acidified water for 30 seconds (0.5% glacial acetic acid diluted in distilled water) before being dehydrated in alcohol and xylene and finally covered with mounting media and cover glass. Picrosirius red stained histological sections were imaged using a polarized light microscope (Leica DM500 microscope, USA). Images of specific areas were captured before and after a 90˚ polarization stage rotation, ensuring consistent imaging conditions, including exposure time and gain.

### Image analysis

PSR image quantification was performed by analyzing the color composition of each image using RGB values. The identification of different colors was based on specific ranges of RGB combinations. Red pixels were defined as having RGB values between (50,0,0) and (255,0,0), yellow pixels were defined as having RGB values between (50,50,0) and (255,255,0), green pixels were defined as having RGB values between (0,50,0) and (0,255,0). Blue pixels were defined as having RGB values between (0,0,50) and (0,0,255). It is important to note that the selection of red and yellow colors was not deliberate but based on the colors in the images. Consequently, only red and yellow pixel values were displayed, while green and blue values were observed to be zero. To determine the color ratios, the number of pixels corresponding to each color (red, yellow, green, and blue) was divided by the total sum of red, yellow, green, and blue pixels. This calculation allowed for the calculation of the percentage color ratio. Each image was processed using Python 3.10 within the PyCharm community version virtual environment. The analysis utilized various modules, including Skimage, NumPy, Pandas, and OpenCV.

### AFM imaging and force spectroscopy

Two unfixed and unstained sections were allocated for AFM-based quantitative nanohistology (QNH). The AFM images and force-distance curves (nano-indentation) were acquired using a

Nano-wizard 4 Bioscience instrument (Bruker, Germany) operating in contact mode under ambient conditions. The imaging and force spectroscopy measurements were performed directly on histological sections within the reticular dermis area of the skin biopsies in ambient conditions. The slides were not rehydrated post-sectioning to avoid further alterations in the collagen structure. MSLN-10-C (k = 0.01N/m) and RTESPA-150 (k = 6N/m) probes (Bruker, Germany) were used for imaging and force spectroscopy, respectively. To correlate the AFM data with the picrosirius red stain images, seven healthy areas marked as red and seven damaged regions marked as greenish yellow in SR stained were selected for imaging and force spectroscopy analysis. The imaging was captured using a size of 5×5 μm, and the settings were optimized for the best results at a frequency of 2.5 Hz. As discussed previously, seven healthy and seven disrupted areas were selected for the indentation measurements. An image was acquired in contact mode at a frequency of 2.5 Hz. Subsequently, a grid of 10×10 was defined on acquired AFM images, and a total of 700 force curves was recorded for each of the healthy and damaged areas of the skin biopsies, providing a comprehensive dataset for analysis. The Young Modulus of the collagen fibrils (E), measured here as the compressive elastic modulus, was determined by fitting the Hertzian model to the force-distance curve acquired from all measurements. The obtained modulus values were then plotted as a distribution to calculate the median and error for all groups. All images were processed, and data analysis was conducted using the JPK data processing software (version 6.3.5). It is important to note that to accurately position the AFM probe on desired areas, two adjacent sections from skin biopsies were used. One was stained with PSR, and the other was unstained and unfixed. Tissue gross morphology was used to align the PSR and optical images on the AFM instrument. Zooming in on the respective images enhanced the geolocation accuracy for the probe placement within the desired regions of interest, however small these may be, even though the AFM data acquisition was performed on an unstained and unfixed section. Since the AFM is equipped with x-y scanner that can place the probe anywhere on the sample surface with high-accuracy (<110nm), it is therefore possible to target the area to be imaged or mechanically probe with a high degree of confidence.

## Kappa-analysis

To investigate whether nanoscale changes in collagen structure could serve as a distinctive marker or fingerprint for EDS, we performed Kappa analysis using nine raters who did not have prior knowledge of AFM imaging or collagen fibril anisotropy interpretation. The assessors were first trained to recognize collagen fibril clarity, D-banding, orientation, and linearity on 3 AFM images of healthy skin. The training emphasized the structure of healthy collagenous tissue involving the four mentioned parameters. The same procedures were followed for 3 AFM images of disrupted collagen. After training, nine assessors were presented with 32 randomized AFM images (16 from each healthy and diseased group). We performed reliability and repeatability tests using Fleiss's and Cohen's kappa score analyses.

**Data statistical analysis.** Kolmogorov–Smirnov test at 0.05 level was used to find significance between all groups. All statistical analyses and data plotting were performed with OriginPro 2021 software.

## Results and discussion

### Histology images analysis

Histological analysis is routinely used to assess structural changes in tissues, including skin, to prognose or diagnose pathologies such as EDS. In Fig 1A, the histological sections stained with H&E show the differences in the quality and orientation of collagen bundles in the skin

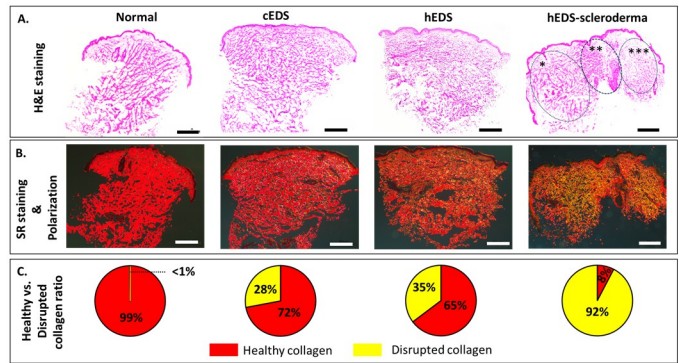

**Fig 1. The histological sections of control, cEDS, hEDS, and hEDS-Scleroderma patients stained with H&E and PS.** (A). H&E-stained histological sections showing thick, organized collagen bundles for control skin and disordered, thin ones in the disrupted regions. The areas marked as *, **, and *** show the heterogeneous collagen density throughout the section for the sample obtained from hEDS, Scleroderma patient. (B). Histological sections from all four groups were stained with PS and imaged under polarized. Control samples showed red stains throughout the section, whereas cEDS and hEDS exhibited localized islets of yellow stained area. In the case of hEDS-Scleroderma, the yellow stain is widespread throughout the section. (C). Pie-chart distribution of the yellow/red stained areas ratio for each patient. Scale bars are 500μm.

samples from healthy individuals and those with EDS. In control tissue, collagen bundles were thick and well-organized compared to the disordered orientation of thin collagen bundles in the other three samples. The control, cEDS, and hEDS present a homogenous collagen density through the dermis. In the case of the hEDS-Scleroderma, we can detect heterogeneity in collagen density with three distinct regions, as labeled in Fig 1. The first region in the dermis highlighted by * appears to be the same density in the three other cases presented. However, two more areas denoted by ** and *** are distinct. The central region in the section (**) presents a highly dense collagen network expanding from the deep reticular dermis up to the dermo-epidermal junction. Likely, this region is directly associated with the scleroderma diagnostics for this patient [23]. The region on the left of the section denoted by (***) appears poorly stained for collagen, presenting a very loose network of collagen fibrils. This structural transition between the highly dense collagen region (**) and collagen-depleted region (***) in the skin would impact the mechanical behavior of the skin [39]. In Fig 1B, we employed polarized light to enhance the contrast in the images stained with PS. After rotating the polarization stage to 90˚, a consistent black background was observed across all images. Collagen bundles, integral components of skin structure, appeared red in the sections obtained from control skin. However, the homogeneous red staining case was not observed in specimens from individuals with EDS. The cEDS section presented small, localized, and discrete islets of yellow-stain collagen throughout the dermis (both in the papillary and reticular dermis). These islets became more apparent and extensive for the hEDS patient. As for the cEDS patient, the islets were randomly distributed throughout the thickness of the dermis. In the case of the hEDS-scleroderma section, the yellow stain is no longer localized as it is spread throughout the section, suggesting that the alterations to the collagen matrix are systemic rather than localized. Interestingly, the three regions of different collagen densities highlighted by the H&E staining did not present major differences in the SR staining when the percentage of red and yellow stained areas was measured. However, it is important to understand that more skin biopsies are needed to correlate the qualitative differences in the areas marked by asterisks in the H&E stain with quantitative analysis in SR stain samples. Overall, the highest concentration of yellow stain was observed in the EDS-scleroderma cases. In addition to this histological

assessment, we used SR staining to explore the alterations in the dermal collagen organization. This Sirus Red dye carries a strong negative charge and interacts with the cationic collagen fibrils due to sulfonic acid groups in SR. This interaction leads to the alignment of SR molecules along the long axis of collagen fibrils [40]. Combining this staining along the length of the fibrils with the natural birefringence properties of collagen associated with its fibrillar D-Banding periodicity, it is possible to combine SR staining with light polarization to observe structural alterations in the dermal collagen matrix organization. Any departure from the characteristic red hue in the polarized images suggests a disruption in the localized alignment of collagen fibrils. Such disruptions could signify irregularities in the structural integrity of the collagen scaffold, as previously shown by other studies [32, 40, 41]. A sustained red hue suggests that the collagen fibrils are arranged in well-defined bundles or sheets, in which the collagen fibrils follow a preferential alignment. A yellow hue indicates a change in the birefringence properties of collagen, suggesting that the collagen fibrils no longer follow a preferential alignment. A green hue would signify that the birefringence property of collagen fibrils is lost (complementary hue to red) and would imply that there is no longer any alignment among the fibrils and that the fibrils are randomly orientated in these regions. Considering that collagen fibrils in a healthy dermis are typically very well aligned into large, dense collagen sheets [33], it is, therefore, possible to understand the level of disruption in the dermal collagen by quantifying the amount of Red/Yellow hue in the sections. Fig1C illustrates the calculated yellow-to-red stain ratio, revealing that in the cEDs section, 72% of the collagen could be considered aligned, with 28% disrupted. The proportion of disrupted fibrils organization increases for the hEDs case (65%) and hEDS scleroderma (92%).

## Nanoscale and histological images correlation

To explore the structural characteristics of the collagen scaffold within the dermal layer, we correlated AFM images site with a polarized image of SR-stained skin sections to identify both intact and compromised regions within the skin. Fig 2A showcases AFM images captured

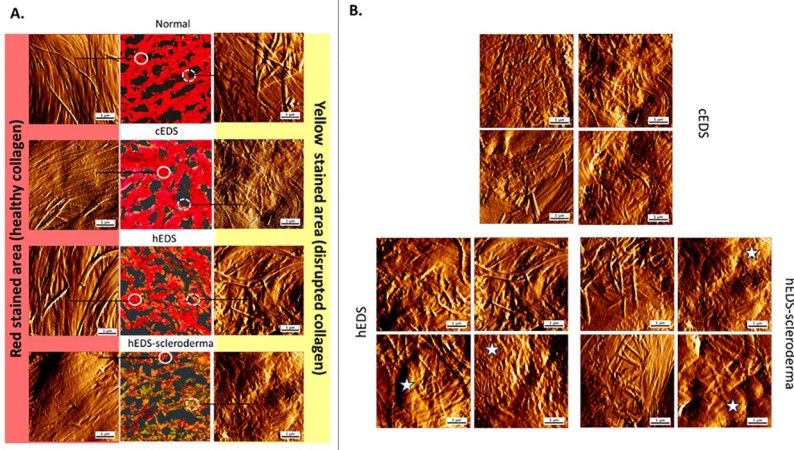

**Fig 2. Correlated AFM and PS-stained images, identifying intact and compromised regions within the skin samples.** (A) showcases selected AFM images from areas of PS-stained samples referred to as red on the right and yellow on the left. All fibrils in the red-stain area of all four cases were structurally homogeneous with long, aligned, and unidirectional collagen fibrils. In contrast, yellow-stained, disrupted areas exhibited random orientation and structural disorganization in the collagen matrix. (B)presents Complementary AFM images from the disrupted area (yellow stained) from cEDS, hEDS, and hEDS-Scleroderma samples, highlighting the random orientation of the collagen fibrils along with wrinkled, swollen, and amorphous collagen matrix (highlighted with stars).

from areas of PS-stained samples denoted as red on the right and yellow on the left. As expected, the red stain area presents some well-defined fibrils collagen organized in a preferential unidirectional alignment. We also use the presence of the D-banding periodicity on the surface of the fibrils to assess whether the fibrils are well-defined. Our previous research has demonstrated that fibrils without D-banding periodicity could be associated with disease phenotype or improperly formed fibrils [32, 42]. In the case of the red stain area, all collagen fibrils presented a recognizable D-banding periodicity. Another parameter to consider is the homogeneity of the fibrillar structure. Collagen fibrils are long and regular cylindrical protein ultrastructure. Any localized alterations in the morphology of the fibrils have been attributed to signs of damage or denaturation. Such localized alterations in the morphology of the fibrils are not usually seen in healthy dermal collagen. Indeed, all fibrils in the red-stain area of all four cases are structurally homogeneous and do not present any notable alterations. However, collagen fibrils' structural homogeneity and unidirectional orientation are challenged in the samples' yellow stain regions, as presented in Fig 2B. While collagen fibrils could still be observed in all disease sections, they did not follow any preferential orientation and presented a level of structural disorganization. This structural disorganization can be characterized by bent or twisted fibrils, which in some present no apparent D-banding. In addition, a specific structural feature of the hEDs and hEDS-scleroderma dermal collagen is the presence of amorphous collagen fibrils or matrix. This amorphous collagen is characterized by swollen non-fibrillar collagen adjacent to healthier collagen fibrils. This amorphous collagen (highlighted by the star in Fig 2B) is reminiscent of damaged collagen undergoing denaturation by hydrolysis [31, 43]. The swollen fibrils and wrinkles in the amorphous collagen matrix suggest that these alterations are not just confined to a single collagen fibril/bundle but that entire areas of the dermis may be affected. This confirms our finding observed by the SR staining in which we observed an increase % prevalence of the yellow stain area for the hEDS and hEDS-Scleroderma cases. The etiology of this amorphous collagen in the dermis is unclear, but its presence would enable the skin hyper-extensibility as seen in patients with hEDS. The presence of sizeable amorphous areas would compromise the scaffolding function of the dermis as dermal collagen relies on its ultrastructural arrangement of parallel fibrils to confer the skin with its unique tensile properties.

### Nanoscale collagen alterations: A fingerprint for EDS?

The interpretation of images, as presented in Fig 2, can be challenging for non-specialists. To explore whether the nanoscale alterations of collagen structure could be used as a fingerprint for EDS, we performed reliability and repeatability tests using Fleiss's and Cohen's kappa score analysis. For this study, we recruited clinicians and biomedical scientists without prior knowledge of AFM or Ehlers-Danlos Syndrome. Following training using a small set of images, nine raters were invited to assign 32 randomized AFM images to either a healthy (red) or disrupted/disrupted (yellow) group. The analysis yielded K = 0.96, and the intra-examiner assessment yielded K = 0.79± 0.1. According to the literature, a Kappa score of 0.8 or more shows excellent agreement between raters. This result confirms that nanoscale dermal collagen structural variations exist between healthy (red) areas and those disrupted or disrupted (yellow) to the extent that even non-specialists can recognize them. However, this approach does not account for subtle variations between the cEDS, hEDS, and hEDS-scleroderma phenotype. To do so would require more patients so that the dataset can be increased and analyzed using a convolutional neural network, for example. Similarly, the reliability of computer tomography angiography-based brain injury assessment was assessed using kappa analysis [44]. Examining the dermal collagen network for structural irregularities in the context of connective tissue

disorders is a promising avenue for research, and it could significantly improve the utility of AFM nanometrology in disease diagnosis. However, combining the %prevalence of the disrupted collagen in a section with the nanoscale assessment of dermal collagen fibrils and matrix could serve as a valuable instrument for assessing the severity of EDS within collagen-rich tissues and hopefully support a novel diagnostics approach for hEDS.

## Collagen fibrils elasticity to determine EDS phenotype?

Although some clinical manifestations of EDS on tissues are well established, little is known about the impact of EDS biomechanical properties on the collagen matrix at the nanoscale. As suggested by Reis [45], "To understand the mechanical complexity of an organ or a complete functional tissue, one must first consider its smallest [structural] constituents, which is the collagen fibrils for the dermis." We performed localized AFM-based nanoindentation directly on individual collagen fibrils in the PS-red stained (healthy collagen) or PS-yellow stained (disrupted collagen) areas to measure Young's Modulus, as presented in Fig 3A & 3B, respectively. In the case of the control case, Young's Modulus of the collagen fibrils follows a unimodal distribution (peak 2) regardless of whether the collagen is from a healthy or disrupted area. As presented in Fig 3B, the Young's moduli of the collagen fibrils recorded on healthy skin agree with our previously reported values for collagen fibrils in dermal collagen. For the cEDS case, we can readily observe the presence of a bimodal distribution of Young's Moduli in both regions. The main peak (peak 1) in both distribution is located at a lower modulus value than that found in our healthy control (Fig 3A). This suggests that the majority of the fibrils in cEDS exhibits a lower compressive Young's modulus peak when compared to the healthy control. Note that a smaller peak (peak 1) can be found at the same modulus value corresponding to the main peak as found in the healthy control. This would mean that a small portion of fibrils still present the same compressive Young's modulus as found in our control sample. For the hEDS case, we can again observe a clear difference between the distribution of Young Moduli for the healthy and disrupted collagen. We recorded a unimodal distribution for the healthy collagen whereas a bimodal distribution was recorded for the disrupted collagen. In this bimodal distribution, the main peak (peak 1) matches that of cEDS but we can also denote

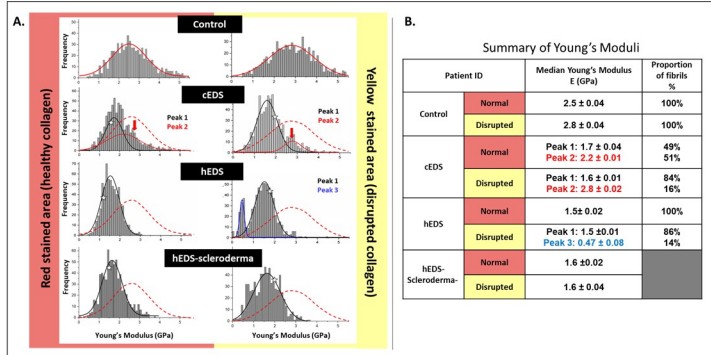

**Fig 3. Collagen fibrils' Young's moduli distributions of skin section's healthy and disrupted area. (A).** Shows an unimodal distribution of Young modulus for control skin in healthy and disrupted regions. This peak shifts gradually to the smaller value, indicating weaker collagen when transiting from control samples to cEDS (indicated by a star). This shift is more pronounced when moving to hEDS and hEDS scleroderma samples. The red dotted line presents the healthy collagen distribution in the control group. **(B).** Summary table of the collagen Young's moduli medians and their prevenances. At 0.05 level, all Younge modulus distributions between healthy and disrupted collagen are significant. Additionally, disrupted and healthy distribution within the same group is significant except for the control skin.

a smaller population (peak 3). The presence of this smaller population with a very low compressive Young's modulus when compared to the healthy control or cEDS is specific to the hEDS patient. Mechanically, the presence of such weak collagen fibrils is noteworthy and may be associated with an increased fragility of the dermal matrix. Finally, in the case of the hEDS-Scleroderma, we can also record a similar fibrillar Young's Moduli distribution population in both healthy and disrupted regions as found for both the cEDS and hEDS cases. Additionally, we have plotted the Young's moduli distribution for all the three regions marked as (\*, \*\*, \*\*) and found similar distribution in all the three regions as shown in S1 Fig, more information on how S1 Fig data was obtained can be found in S1 File. This mechanical data must be interpreted carefully as we have only 4 cases in the present study. However, each histogram in Fig 3A & 3B contains 700 indentations recorded over seven distinct areas. This ensures that the data presented is representative of each case. The trends observed in the mechanical findings will need to be cross-validated with more cases. Still, the early findings from this mechanical study offer new hope for finding specific biomarkers for hEDS as, to date, hEDS does not have specific structural or biomechanical fingerprints[22], preventing or delaying accurate diagnosis and therapy. While diagnosing cEDS is rapid (via genetic testing), it takes an average of ten to 16 years from the onset of symptoms before a correct diagnosis of hEDS is made. A lack of timely and confirmed diagnosis impacts patients' health insurance supporting adequate care. Inaccurate diagnosis also increases the emotional burden of disease for patients and caregivers. These unmet clinical needs further contribute to morbidity and patients' feelings of invalidation and global dissatisfaction [46].

## There is a need for more diagnostic tools for hEDS

Establishing an accurate and timely diagnosis of all EDS types is a fundamental clinical management problem. The path to an EDS diagnosis starts with an examination, review of the patient's medical history, and physical testing that includes a) An assessment of joint mobility using the Beighton Scale [9], b) Visual assessment of abnormal scarring and skin testing to determine the extent of stretching. Establishing a definitive diagnosis for all the EDS subtypes, except for hEDS, also involves confirmation by genetic testing to identify the affected gene in each subtype. Failure to provide early detection and accurate diagnosis leads to treatment errors and ineffective preventive approaches to ameliorate recognized symptoms, disabilities, and complications. Before 2017, hEDS was diagnosed based on the Villefranche nosology (framework for categorizing EDS [9]), which needed more well-defined inclusion and exclusion criteria. hEDS diagnosis has primarily involved subjective interpretation, which reduces diagnostic accuracy [9]. The 2017 diagnostic criteria for hEDS were developed to define better the EDS spectrum (particularly hEDS) and to differentiate it from other pathologically distinct clinical conditions. Further, the 2017 diagnostic criteria for hEDS were reliant on the development of a comprehensive physical exam template, which involved the review of clinical manifestations in three categories: (1) Presence of generalized joint hypermobility; (2) systemic manifestations, musculoskeletal complications, and family history; (3) absence of other CTDs [9]. These criteria were found challenging to apply since the phenotypic presentation of patients with EDS is so diverse that a "one-size-fits-all" diagnostic template is not tractable [9]. To improve diagnosis, we proposed updates to the current diagnostic criteria using a cumulative scoring system based on a weighted average of symptoms in the order of importance. But despite these suggestions for improved phenotyping, hEDS is still diagnosed using specific signs and symptoms, which are open to subjective interpretation by physicians, leading to diagnostic errors and inappropriate treatments [16]. In EDS, fibrillar collagens are the predominantly targeted proteins and abnormal fibrillar structures are seen. Yet, quantifying the

structural and functional impact of this condition on collagen at the sub-micron level remains challenging [2, 14, 47]. This is due to a) a need for current techniques to assess these properties systematically at the nanoscale and b) a lack of histopathology standards (at that scale) in the literature. One promising diagnostic approach is based on the structural examination of skin biopsies using transmission electron microscopy. This approach could be used as a diagnostic tool and is often combined with a molecular diagnosis of EDS (but only when feasible). Several abnormal ultrastructural characteristics indicate a monogenic subtype-like classical or derma-tosparaxis subtype in EDS [48, 49]. While exploring the cross-section of collagen fibrils using transmission electron microscopy has helped understand EDS etiology, the quantification of collagen EDS-related morphometric variations along the long axis of the fibrils is found to be lacking. With the translation of imaging techniques such as AFM as part of a histological approach, it is now possible to characterize the fibrils along their long-axis, as done in this study. More puzzling has been the lack of reports on measuring the mechanical properties of individual collagen fibrils in EDS. Again here, the AFM excels in this task and will provide new knowledge (albeit at the fibrils scale) that can be used to create a collagen-based pheno-type diagnostic tool. Thus, the relationship between form, function, and the structural prop-erty of collagen remains to be defined in EDS at all relevant scales, ranging from the collagen fibril to the entire tissue. By employing tools such as the Atomic Force Microscope, we can start cataloging the complexity of the dermal matrix at the nanoscale as found in EDS, as we have done in this study.

## Summary table of key findings

Table 1 summarizes this study's key findings and can be used as a guiding chart for future studies exploring the EDS dermal collagen phenotype. Although the outcomes presented in this table are specific to the case series showcased in this study, they can serve as a guide for future population studies to evaluate whether these nanoscale morphometric and mechanical properties of dermal collagen can be used to support the diagnosis of hEDS.

**Table 1. Summary table of key findings.**

| | | | Control | cEDS | hEDS | hEDS-Scleroderma |
|---|---|---|---|---|---|---|
| **Tissue-scale histology** | **H & E** | | Thick, organized Homogenous density | Thin, disordered Homogenous density | Thin, disordered Homogenous density | Thin, disordered Heterogenous density |
| | **SR & Polarization** | | Homogenous red stain Red stained areas: 100% Yellow stained areas: <1% | Discrete, small yellow islets Red stained areas ~ 70% Yellow stained areas ~ 30% | Discrete, larger yellow islets Red stained areas ~ 65% Yellow stained areas ~ 35% | Widespread yellow area Red stained areas <10% Yellow stained areas >90% |
| **Nanoscale histology** | **Imaging** (collagen fibrils) | **Normal area** | Long, parallel alignment, D-Banding apparent | Long, parallel alignment, D-Banding apparent | Long, parallel alignment, D-Banding apparent | Long, parallel alignment, D-Banding apparent |
| | | **Disrupted area** | Early signs of matrix disorganization (loss of parallel alignment), D-Banding apparent | Complete loss of parallel alignment, D-Banding apparent | Complete loss of parallel alignment, Strong presence of amorphous collagen fibrils/matri | Complete loss of parallel alignment, Strong presence of amorphous collagen fibrils/ matrix |
| | **Mechanics** (collagen fibrils) | **Normal area** | Unimodal distribution ($E \sim 2.5$ GPa) | Bimodal distributions with the $2^{nd}$ distribution towards lower E. | Unimodal distribution With low E (~1GPa lower than control) | Unimodal distribution With low E (~1GPa lower than control). Possible subpopulations at lower E |
| | | **Disrupted area** | Same unimodal distribution as found in normal | Bimodal distributions with very prominent $2^{nd}$ distribution towards lower E. | Bimodal distributions with the $2^{nd}$ distribution towards even lower E than normal. | Unimodal distribution With low E (~1GPa lower than control). |

## Conclusion

In conclusion, histology image analysis combined with nanoscale investigations using Atomic Force Microscopy (AFM) has provided valuable insights into the structural alterations within the dermal collagen matrix in individuals with Ehlers-Danlos Syndrome (EDS). These findings could revolutionize the diagnosis and understanding of EDS subtypes, particularly hypermobile EDS (hEDS), which often lacks specific diagnostic markers. Histological analysis revealed distinct differences in collagen organization and density between healthy individuals and those with EDS, with a notable impact on the mechanical behavior of the skin. Polarized light microscopy further emphasized these differences, highlighting the presence of disrupted collagen in EDS cases. Additionally, the use of Sirius Red staining and polarization allowed for quantifying collagen disruption, providing valuable information for assessing EDS severity. The correlation between nanoscale AFM images and histological staining patterns demonstrated unique structural characteristics within the collagen scaffold of EDS patients. These characteristics, such as disrupted collagen fibrils and amorphous collagen, could be potential biomarkers for EDS subtypes. The reliability and repeatability of identifying disrupted collagen regions by non-specialists through AFM-based nanoscale assessments further support the potential for developing diagnostic tools based on collagen structural variations. Furthermore, evaluating collagen fibril elasticity using AFM nanoindentation revealed distinct mechanical properties in healthy and disrupted collagen regions. These findings can potentially contribute to developing specific biomechanical markers for EDS subtypes, particularly hEDS, which currently lacks diagnostic criteria.

## Supporting information

**S1 File. Analyzing regions marked by (\*, \*\*, \*\*\*) on hEDS-Scleroderma skin sample.** (DOCX)

**S1 Fig. Shows the distributions of Young's moduli of collagen fibrils recorded in the \*, \*\*, and \*\* regions of the hEDS-Scleroderma case.** A unimodal distribution of Young's moduli was observed for both normal and disrupted collagen across all three regions. The red dotted line represents the distribution of healthy collagen in the control group. (TIF)

## Acknowledgments

The authors would like to acknowledge the participation of the EDS patients from the UHN GoodHope EDS clinic (University Health Network, Toronto, Canada) and its Director, Dr. Hance Clarke, for supporting the project. The authors would also like to acknowledge Dr. Grahame, who suggested this study in 2010. The authors would finally like to thank the CAMiLoD Facility at the Faculty of Dentistry, University of Toronto, for the extensive access to their facility.

## Author Contributions

**Conceptualization:** Nimish Mittal, Laurent Bozec.

**Data curation:** Mehrnoosh Neshatian, Laurent Bozec.

**Formal analysis:** Mehrnoosh Neshatian, Sophia Huang, Aiman Ali, Laurent Bozec.

**Funding acquisition:** Nimish Mittal, Laurent Bozec.

**Investigation:** Mehrnoosh Neshatian, Nimish Mittal, Laurent Bozec.

**Methodology:** Mehrnoosh Neshatian, Aiman Ali, Emilie Khattignavong, Laurent Bozec.

**Project administration:** Laurent Bozec.

**Resources:** Nimish Mittal, Laurent Bozec.

**Software:** Mehrnoosh Neshatian, Sophia Huang, Emilie Khattignavong, Laurent Bozec.

**Supervision:** Nimish Mittal, Laurent Bozec.

**Validation:** Mehrnoosh Neshatian, Laurent Bozec.

**Visualization:** Mehrnoosh Neshatian, Laurent Bozec.

**Writing – original draft:** Mehrnoosh Neshatian, Laurent Bozec.

**Writing – review & editing:** Mehrnoosh Neshatian, Nimish Mittal, Sophia Huang, Aiman Ali, Emilie Khattignavong, Laurent Bozec.

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
