## [Decision Letter · Decision Letter 0]

5 Jun 2024

PONE-D-24-12071Investigation of dermal collagen nanostructures in Ehlers-Danlos Syndrome (EDS) patients.PLOS ONE

Dear Dr. Bozec,

Thank you for submitting your manuscript to PLOS ONE. After careful consideration, we feel that it has merit but does not fully meet PLOS ONE’s publication criteria as it currently stands. Therefore, we invite you to submit a revised version of the manuscript that addresses the points raised during the review process. Please address each of the comments below from the reviewer. I strongly recommend that you make appropriate changes as suggested by the reviewer.

We look forward to receiving your revised manuscript.

Kind regards,

Richard G. Haverkamp, PhD

Academic Editor

PLOS ONE

“1.GoodHope Ehlers-Danlos Syndrome Foundation

2.Rosenstadt Fund, University of Toronto.”

“The authors would like to acknowledge the participation of the EDS patients from the UHN GoodHope EDS clinic (University Health Network, Toronto, Canada) and its Director, Dr. Hance Clarke, for supporting the project. The authors would also like to acknowledge Dr. Grahame, who suggested this study in 2010. Partial funding was received from the UHN GoodHope EDS clinic and the Faculty of Dentistry Bridge Grant program to support the study. The authors would finally like to thank the CAMiLoD Facility at the Faculty of Dentistry, University of Toronto, for the extensive access to their facility. The authors declare no conflict of interest in the work presented.”

“1.GoodHope Ehlers-Danlos Syndrome Foundation

2.Rosenstadt Fund, University of Toronto.”

5. Please remove your figures from within your manuscript file, leaving only the individual TIFF/EPS image files, uploaded separately. These will be automatically included in the reviewers’ PDF.

Additional Editor Comments:

I am sorry we have only one reviewer for your manuscript, but in the interests of time I have decided to proceed with a decision. I also have read the manuscript and feel the manuscript is suitable for publication after the points described are addressed.

Reviewers' comments:

Reviewer's Responses to Questions

**Comments to the Author**

1. Is the manuscript technically sound, and do the data support the conclusions?

Reviewer #1: Yes

2. Has the statistical analysis been performed appropriately and rigorously? 

Reviewer #1: Yes

3. Have the authors made all data underlying the findings in their manuscript fully available?

Reviewer #1: Yes

4. Is the manuscript presented in an intelligible fashion and written in standard English?

Reviewer #1: Yes

5. Review Comments to the Author

Reviewer #1: It is a well written paper addressing the possibility to use atomic force microscopy as an adjunct for clinical diagnosis of Ehlers Danlos.

A have a number of comments:

1. The authors describe the use of genetic testing, including EDS related gene panel, in the studied individuals.

Please provide the list of genes included in the aforementioned panel and include the molecular details and ACMG classification of the COL1A1 variant identified in the cEDS case.

2. The authors show skin biopsy of the hEDS-scleroderma patient to display patchy abnormalities in terms of collagen fiber density with three different regions that could be identified by light microscopy,, while the one with the highest fiber density was ascribed to scleroderma related changes. It seems prudent to analyze the AFM findings including collagen fiber elasticity measurements separately for each one of these three regions of the hEDS-scleroderma patient.

This seems important since previous paper utilizing AFM including Young’s Modulus measurements in the setup of scleroderma showed increased collagen fiber stiffness (PMID 28138238). This point should be discussed.

3. The details of full clinical and laboratory findings of the hEDS-scleroderma patient should be included, with the emphasis on the clinical scleroderma subtype, autoantibodies present and disease duration.

4. The existing literature addressing the previous use of AFM for connective tissue disorders should be included in introduction.

5. minor comments:

page 5 line 79 - "genetic variants of EDS" while implying additional genes causative of EDS. Please rephrase

page 5 line 85 - typo, "relayed" should be "related"

6. PLOS authors have the option to publish the peer review history of their article (what does this mean?). If published, this will include your full peer review and any attached files.

Reviewer #1: No

---

## [Author Response · Author response to Decision Letter 0]

3 Jul 2024

To the Editor and the Reviewer,

The research team would like to acknowledge and thank the two reviewers for reviewing our manuscript and providing us with constructive feedback, which has been addressed below:

1. The authors describe the use of genetic testing, including EDS related gene panel, in the studied individuals.

Please provide the list of genes included in the aforementioned panel and include the molecular details and ACMG classification of the COL1A1 variant identified in the cEDS case.

Thank you for your comment. The standard EDS related comprehensive gene panel was done at our institute employs Next generation Sequencing (NGS) with duplication and deletion analysis by exon targeted microarray. This test includes 22 genes. Details can be found online on the website (https://www.sickkids.ca/siteassets/care--services/for-health-care-providers/lab-information-sheets/connective-tissue-disorders-ehlers-danlos-syndrome.pdf). In the cEDS case, patient was found to have a pathogenic variant (c.1053delG in exon7) of the COL5A1 gene.

The body of the manuscript has been edited accordingly (as follow). 

Section:

Clinical cases: Next generation Sequencing (NGS) with duplication and deletion analysis by exon targeted microarray was performed at SickKids hospital for diagnosis. This test includes 22 genes. Details can be found online on the SickKids website.

Study ID No 5 (labeled as cEDS throughout the paper)

Genetic testing revealed a COL1A1 mutation (c.1053delG in exon7), confirming the diagnosis of classical EDS.

2. The authors show skin biopsy of the hEDS-scleroderma patient to display patchy abnormalities in terms of collagen fiber density with three different regions that could be identified by light microscopy, while the one with the highest fiber density was ascribed to scleroderma related changes. It seems prudent to analyze the AFM findings including collagen fiber elasticity measurements separately for each one of these three regions of the hEDS-scleroderma patient.

This seems important since previous paper utilizing AFM including Young’s Modulus measurements in the setup of scleroderma showed increased collagen fiber stiffness (PMID 28138238). This point should be discussed.

AFM images (5x5 µm) were acquired from three normal and three disrupted areas (guided by PS image) of the histological section of the skin biopsy. Subsequently, a total of 300 force curves were recorded for each of the normal and disrupted areas. A unimodal Yong’s moduli distribution was observed for both normal and disrupted collagen in all the three regions similar to hEDS-Scleroderma curves presented in Figure 3A. Suggesting that in each of these regions, a single population of collagen fibrils is present. These mechanical data must be interpreted carefully, as we only had one case with both EDS and scleroderma. Further association of the mechanical findings and H&E staining requires more skin tissue from similar patients

These findings have been added as supporting information to the manuscript. And the body of the manuscript on the section “Collagen fibrils elasticity to determine EDS phenotype? “, has been edited as follow: “Additionally we have plotted the Yoong’s moduli distribution for all the three regions denoted as (*, **, **) and found similar distribution in all the three regions as shown in S1Fig”

Figure S1 Shows the distributions of Young's moduli of collagen fibrils recorded in the *, **, and ** regions of the hEDS-Scleroderma case. A unimodal distribution of Young's moduli was observed for both normal and disrupted collagen across all three regions. The red dotted line represents the distribution of healthy collagen in the control group

3. The details of full clinical and laboratory findings of the hEDS-scleroderma patient should be included, with the emphasis on the clinical scleroderma subtype, autoantibodies present and disease duration.

We thank you for the suggestion. The focus of this paper is to highlight the molecular histophysiologal profile of collagen fibrils and to keep the manuscript succinct, we had not included significant details of the clinical phenotype. We have made modifications and included the relevant clinical and laboratory findings, duration and autoantibodies. We have edited the body of the manuscript under the section, clinical cases - Study ID No 4 (labeled as hEDS- Scleroderma throughout the paper) “patient had a 10-year history of inflammatory arthralgias involving PIP and MCP joints, diffuse muscle and joint pains affecting multiple areas of body, secondary diagnosis of fibromyalgia, patchy areas of skin thickening, and Raynaud’s phenomenon. History was negative for uveitis, iritis and psoriasis. Lab testing was positive for Anti-Nuclear Antibody (ANA) and Anti centromere Antibody (ACA) and negative for Anti CCP and rheumatoid factor.”

4. The existing literature addressing the previous use of AFM for connective tissue disorders should be included in introduction.

We thank the reviewer for the comment. We have included the following in introduction: 

AFM has previously used for connective tissue morphometry studies such as Study of the Extracellular Connective Tissue Matrix in Patients with Pelvic Organ Prolapse [34], nanohistological investigation of scleroderma [35], oral submucous fibrosis [36], assessments of extracellular matrix in intervertebral disc and degeneration [37], and many other studies. In this study we have employed AFM to examine ex-vivo skin biopsies obtained from EDS patients to characterize collagen morphometric differences in cEDS and hEDS.

5. minor comments:

page 5 line 79 - "genetic variants of EDS" while implying additional genes causative of EDS. Please rephrase

Thank you for the comment. The sentence has been edited as follow “hEDS shares several clinical features with EDS subtypes but lacks confirmed genetic markers”

page 5 line 85 – typo, “relayed” should be “related”

We are sorry for the type. It is now corrected.

---

## [Editor Report · Decision Letter 1]

5 Jul 2024

Investigation of dermal collagen nanostructures in Ehlers-Danlos Syndrome (EDS) patients.

PONE-D-24-12071R1

Dear Dr. Bozec,

We’re pleased to inform you that your manuscript has been judged scientifically suitable for publication and will be formally accepted for publication once it meets all outstanding technical requirements.

Kind regards,

Richard G. Haverkamp, PhD

Academic Editor

PLOS ONE
---

## [Editor Report · Acceptance letter]

12 Jul 2024

PONE-D-24-12071R1 

PLOS ONE

Dear Dr. Bozec, 

I'm pleased to inform you that your manuscript has been deemed suitable for publication in PLOS ONE. Congratulations! Your manuscript is now being handed over to our production team.

Kind regards, 

on behalf of

Professor Richard G. Haverkamp 

Academic Editor

PLOS ONE